# Investigating the Critical Factors of Professionals’ BIM Adoption Behavior Based on the Theory of Planned Behavior

**DOI:** 10.3390/ijerph18063022

**Published:** 2021-03-15

**Authors:** Zezhou Wu, Mingyang Jiang, Heng Li, Xiaochun Luo, Xiaoying Li

**Affiliations:** 1Sino-Australia Joint Research Center in BIM and Smart Construction, Shenzhen University, Shenzhen 518000, China; wuzezhou@szu.edu.cn (Z.W.); jiangmingyang2017@email.szu.edu.cn (M.J.); 2Key Laboratory of Coastal Urban Resilient Infrastructures (MOE), Shenzhen University, Shenzhen 518000, China; 3Department of Building and Real Estate, The Hong Kong Polytechnic University, Hong Kong, China; heng.li@polyu.edu.hk (H.L.); bsericlo@polyu.edu.hk (X.L.)

**Keywords:** building information modeling (BIM), theory of planned behavior, BIM adoption, critical factors, China

## Abstract

In recent years, building information modeling (BIM) has been receiving growing interest from the construction industry of China. Nevertheless, although BIM has many foreseeable advantages, many studies claimed that these advantages have not been sufficiently achieved in practice at the current stage. In this circumstance, it is interesting to investigate what really drives the adoption of BIM. Based on Ajzen’s theory of planned behavior (TPB), a hypothetical model which involves nine latent variables is initially established. Then, a questionnaire is designed and distributed to the construction professionals in the Chinese context. After reliability and validity analysis, the goodness-of-fit of the initial model and the related theoretical assumptions are tested through structural equation modeling (SEM). Based on the modification indicators, a modified model is finally derived. Results show that economic viability and governmental supervision are the most critical factors that influence construction professionals’ BIM adoption behavior in China, sharing weights of 0.37 and 0.34, respectively, whereas other factors play limited roles in this regard. The research findings revealed from this study can provide insightful references for countries that intend to promote BIM adoption in a similar circumstance.

## 1. Introduction

In recent years, informatization has been received substantial emphasis in the construction industry [1,2,3,4,5,6]. Jung et al. [7] stated that information is a critical resource in a construction project, facilitating not only effective project management, but also automation in engineering and construction. Furthermore, Lu et al. [8] argued that a particular building could be viewed as a cluster of information, and information management is critical in the process of construction project management. Nowadays, lean construction and information technologies have been used in different kinds of projects [9,10,11,12]. Under this background, building information modeling (BIM), which is regarded as a revolutionary technology for conducting effective information management during the lifecycle of a construction project, has gained growing interest from both academia and industry.

Currently, there are various definitions of BIM. For example, the National Institute of Building Sciences (NIBS) of the United States specifies that BIM is a business process for generating and leveraging building data to design, construct and operate the building during its lifecycle [13]. According to the National Building Specification (NBS) of the United Kingdom, BIM is regarded as a process for creating and managing information on a construction project across the project lifecycle [14]. Though BIM has been given different literal definitions by different countries and organizations, it is widely regarded as an effective technology for construction management. Currently, the implementation of BIM has become an emerging trend in the construction industry [15,16,17,18,19,20]. The potentials of BIM implementation and its benefits have been studied in existing literature, as shown in Table 1.

However, in practice, despite these “theoretical” advantages, barriers and limitations were also observed during the practical BIM implementation [66,67,68,69,70,71]. In the technical aspect, Enshassi et al. [72] claimed that the data exchange and validation had not been thoroughly investigated. Venugopal et al. [73] suggested a modular and logical framework based on the formal specification of industry foundation classes (IFC) concepts should be developed. In addition, Lee et al. [74] revealed that the mechanism of data exchange standards still faces many challenges. In the non-technical aspect, Abd Jamil and Fathi [75] argued that BIM faces legal and contractual issues during its implementation. Raouf and Al-Ghamdi [76] reviewed BIM implementation in green buildings and found that high upfront costs and delays, design complexities and documentation requirements, superior performance enhancement requirements, and skewness towards environmental sustainability were the four major obstacles. Furthermore, Teng et al. [77] and Yu et al. [78] stated that the unbalanced profit distribution among stakeholders may hinder the active implementation of BIM.

From the above literature review, it can be seen that, as an innovative technology to provide lifecycle management throughout the complex construction process, BIM has shown great potential of solving the problems in the traditional architectural, engineering, and construction (AEC) industry. However, the advantages have not been sufficiently achieved in practice. In this circumstance, it is interesting to investigate what really drives the adoption of BIM.

In recent years, a growing number of studies have been conducted to identify the BIM adoption factors to promote BIM application in real practice [79,80]. Through the combination of the Collaborative BIM Decision Framework and a Focused Group Interviews analysis, Gu and London [69] have divided the driving factors of BIM adoption in the AEC industry into two parts (i.e., the requirement of technical tool functional and issues of non-technical strategies). Using content analysis, Mohammad et al. [81] have systematically investigated the adoption factors of BIM from the previous studies and summarized 24 most frequent variables. A five-step empirical approach has been established to understand the critical success factors (CSF) of BIM adoption at a corporate level. This study found that the “support from top management” and “functionality” of BIM tools are the two most significant factors for BIM adoption among total 58 CSFs [82]. More recently, Ullah et al. [83] also implemented the Technology Organization and Environment framework to investigate and classify the factors that affect BIM adoption in a contemporary public authority from the literature. However, scant research has been provided to characterize the critical factors of BIM adoption behavior from the perspective of the Theory of Planned Behavior (TPB). In this theory, three main variables, such as attitude, subjective norm, and perceived behavioral control, are regarded to determine the intention of performing behavior. It is suitable to act as a basic theoretical framework by combing other external affecting factors, and it has been widely used to explore the innovation adoption behavior in other domains [84,85]. The remainder of this paper is organized as follows. Firstly, an initial theoretical model and eight research hypotheses are proposed based on the summary of existing studies. This is followed by presenting the research methodology employed in this study. Subsequently, structural equation modeling is conducted to identify the drivers of BIM adoption in China and corresponding discussions are presented. This paper ends with a conclusion section.

## 2. Research Hypotheses

Individuals’ behavioral intentions may influence successful BIM implementation. Xu et al. [86] tested individuals’ BIM adoption behavior from three dimensions (i.e., technology dimension, organizational dimension, and attitude dimension), arguing that the attitude dimension could indirectly and positively affect the actual use of BIM by enhancing their interest in learning BIM technology. Howard et al. [87] investigated 84 industry stakeholders from the United Kingdom and found that the attitudes and intentions have direct and positive influences on the individuals’ adoption of BIM. Jin et al. [88] also claimed that practitioners’ perceptions towards BIM could affect its adoption. In this study, the TPB was selected as the basis for measuring intentions. Three variables, such as attitude towards behavior, subjective norm, and perceived behavioral control, are regarded as having direct positive effects on behavioral intentions. Thus, the following hypotheses are proposed.

**Hypotheses** **1 (H1).**
*Attitude towards behavior (AB) has a direct positive effect on behavioral intention (BI).*


**Hypotheses** **2 (H2).**
*Social norm (SN) has a direct positive effect on behavioral intention (BI).*


**Hypotheses** **3 (H3).**
*Perceived behavioral control (PBC) has a direct positive effect on behavioral intention (BI).*


**Hypotheses** **4 (H4).**
*Behavioral intention (BI) has a direct positive effect on BIM adoption behavior (B).*


Technical feasibility is widely considered as an essential factor that affects the application of BIM in the construction industry. Ding et al. [89] found that technical defects and BIM capability are the key factors that hinder the architects’ implementation of BIM. To achieve four-dimensional BIM, Lopez et al. [90] reviewed various technical issues concerning the usability of achieving four-dimensional BIM. Ghaffarianhoseini et al. [91] further claimed that the definitive benefits of BIM had not been adequately capitalized upon due to technical issues. Zou et al. [92] also argued that existing technical limitations (e.g., incompatibility with partners) may cause risks during BIM implementation. Therefore, the following hypothesis is proposed.

**Hypotheses** **5 (H5).**
*Technical feasibility (TF) has a direct positive effect on BIM adoption behavior (B).*


Economic viability may be a significant factor in adopting BIM technology. Cao et al. [93] examined the motives of BIM implementation, revealing that economic motives are significantly associated with the level of BIM adoption. Liao and Teo [94] specified advantages and financial support is a critical success factor of BIM implementation in Singapore. In another study conducted by Cao et al. [48], the importance of economic viability was further confirmed. Lee et al. [95] analyzed the economic feasibility of implementing a structural building information modeling (S-BIM) on high-rise building structures. Saieg, Sotelino, Nascimento and Caiado [41] also discussed the economic aspect of adopting BIM in lean construction. Therefore, the following hypothesis is proposed.

**Hypotheses** **6 (H6).**
*Economic viability (EV) has a direct positive effect on BIM adoption behavior (B).*


The industrial environment may influence the adoption of BIM because coordinating various stakeholders of a project is the main advantage of BIM technology. Porwal and Hewage [96] indicated that a large proportion of clients from the public sector are afraid of using BIM in their projects because they think the market is not ready for BIM. Sacks et al. [97] reviewed fifteen BIM guidelines, and standard and protocol documents and found missing aspects in some of these documents. Papadonikolaki and Wamelink [98] argued that the inter- and intra-organizational conditions are important for integrating BIM with the supply chain. Recent research conducted by Abd Jamil and Fathi [75] stated that there are still many contractual challenges to be solved for BIM-based construction projects. Therefore, the following hypothesis is proposed.

**Hypotheses** **7 (H7).**
*Industrial environment (IE) has a direct positive effect on BIM adoption behavior (B).*


Governmental supervision could affect stakeholders’ adoption of BIM because regulations and policies can determine an organization’s actual behavior. Cheng and Lu [99] examined the efforts made by public sectors and argued that the public sector plays a significant role in promoting BIM in the AEC industry. Chang et al. [100] even suggested that the government can mandatorily require the compulsory adoption of BIM in public projects. In addition, as the intellectual property rights (IPR) in BIM projects is of great concern [101], it is necessary for the government to make relevant regulations to protect different stakeholders’ intellectual property as well as other benefits. Therefore, the following hypothesis is proposed.

**Hypotheses** **8 (H8).**
*Governmental supervision (GS) has a direct positive effect on BIM adoption behavior (B).*


Based on the proposed hypotheses, a preliminary theoretical model was developed, as shown in Figure 1.

## 3. Research Methodology

In this study, a questionnaire was designed to investigate construction professionals’ BIM adoption behavior and the affecting factors. Three sections were involved in the questionnaire. The first section investigated the background information of the respondents, including the working category, gender, education level, the number of projects participated in, etc. The second section presented the measurement scales for the nine latent variables, as shown in Table 2. The questions used for constituting the measurement scales were initially designed according to the basic guidelines of TPB (i.e., AB, SN, PBC, and B) and the items identified from existing literature (including TF, EV, IE, and GS). A five-point Likert scale was employed, ranging from “1” (strongly disagree) to “5” (strongly agree). In the third section, an open question, namely, “please provide any comments on this questionnaire”, was proposed to invite the respondents to provide their opinions. Then, three interviews with two experienced professionals and one scholar whose research interest is in this field were conducted to improve the questionnaire.

The questionnaires were collected by two means. The first way was distributing the questionnaire in BIM-related professional forums. Nevertheless, the collected responses were very limited, only 47 responses were collected. In order to collect more responses, the questionnaires were then sent to construction professionals via email. The “snowball sampling” strategy was adopted by inviting respondents to hand out the questionnaire to their colleagues. This strategy was adopted because it could improve the efficiency of obtaining a relatively large number of responses [102,103]. A total of 244 responses were collected from these two methods. Subsequently, a filtering process was conducted to screen invalid questionnaires. After the filtering process, 206 valid responses were left for further analysis, representing 84.4% of the total responses. Structural equation modeling (SEM) was used as the main technique for the analysis. SEM was selected because it is a recognized method for testing hypotheses with empirical data [104]. Confirmatory factor analysis was conducted to test the validity of the measurement models and path analysis was employed to test the goodness-of-fit of the proposed model. After obtaining the optimized model, the significant influencing factors and corresponding regression weights can be determined. Discussions were further made by interviews with three experienced professionals.

## 4. Results

### 4.1. Descriptive Statistics

The descriptive statistics of the 206 respondents were analyzed using SPSS, as shown in Figure 2. From Figure 2, it can be seen that the respondents were mainly from the developer, contractor, and research institution, representing 85.9% in total. In addition, most of the respondents had relatively less working experience, a total of 87.9% of the respondents had working experiences less than 10 years. More than 64% of the respondents had a bachelor’s degree and 35.4% of the respondents had a master’s degree or above. Similar to the short working period, the number of participated projects was relatively limited, more than 90% of the respondents participated in less than 10 projects.

### 4.2. Reliability and Validity Analysis

Reliability analysis was conducted to test the consistency of the measurement items. Cronbach’s α coefficient was checked for the measurement scales of each latent variable and the whole questionnaire. The value for the Cronbach’s α coefficient ranges from 0 to 1, the reliability of the measurement scales is considered to be high if the α value is greater than 0.8. In this study, the Cronbach’s α coefficients of the latent variables of AB, SN, PBC, BI, TF, EV, IE, GS, and B were 0.901, 0.896, 0.927, 0.925, 0.896, 0.899, 0.894, 0.940, and 0.932, respectively, as shown in Table 3. In addition, the reliability coefficient of the whole questionnaire was 0.906, indicating that the reliability of the questionnaire is good and can be further tested for validity.

Factor analysis was employed to test the validity. From Table 3, it can be seen that all Kaiser-Meyer-Olkin (KMO) values exceeded the recommended good level of 0.7, and all Bartlett test of sphericity in the questionnaire maintained at the level of 0.05. In addition, all the factor loadings were between 0.743 and 0.943, exceeding the acceptable level of 0.5. The results indicate that all latent variables passed the validity test and can be analyzed in the next step of the structural equation model.

### 4.3. Structural Equation Modeling

#### 4.3.1. Establishing the Initial Model

According to the preliminary theoretical model proposed in Figure 1, an initial structural equation model was established by using AMOS 24.0, as shown in Figure 3. In the initial model, there are 9 latent variables and 44 observed variables. As there may be calculation errors in the data fitting process, a total of 44 error terms of observation variables and 2 error terms of latent variables were set in the model, which are numbered e1–e44 and u1–u2. The number of distinct sample moments was 990 and the number of distinct parameters to be estimated was 123. Therefore, the degree of freedom of the default model was 867, which means the model is identifiable. As all of the factor loadings were higher than 0.5, no observed variable needs to be deleted.

#### 4.3.2. Model Fitting of the Initial Model

Considering that the model in this paper has a multivariate estimation program of 9 latent variables and 44 observation variables, the skewness and kurtosis of each item of the scale are less than 2, and the sample data conform to normal distribution. Thus, the maximum likelihood method was selected for model estimation. The path coefficient of the initial model is shown in Table 4. From Table 4, it can be seen that several paths existed with insignificant *p*-values.

Similar to Cronbach’s α, construct reliability (CR) as a reliability index for testing latent variables. The higher the value of the CR, the higher the consistency of the internal consistency, and the 0.7 is an acceptable threshold [105]. The average of variance extracted (AVE) was also calculated. AVE calculates the explanatory power of variation of latent variables. The higher AVE is, the greater the percentage of variation of indicator variables explained by latent variables will be. Fornell and Larcker [105] suggested that the ideal value should be greater than 0.5.

The composition reliability and average of variance extracted of the nine latent variables can be calculated as shown in Table 5.

The goodness of fit of the initial model is shown in Table 6. From Table 6, most of the goodness-of-fit indices satisfy their corresponding acceptable requirements. However, the measures of AGFI (Adjusted Goodness of Fit), NFI (Normed Fit Index), and RFI (Relative Fit Index) could not pass the acceptance level, indicating that the model needs to be optimized.

#### 4.3.3. Model Modification

According to the path coefficient derived from the initial model testing, the *p*-value of the path from TF to B is 0.850 (see Table 4), indicating that the path is not significant [106]. Similar analysis procedures were conducted with the updated model. The derived results showed that the *p*-value of the path from IE to B was 0.632, which was still not significant. Therefore, the hypothesis of H7 was rejected. After several rounds of modification, the paths PBC → BI, BI → B were subsequently deleted, and the modified final structural model was derived, as shown in Figure 4.

The path coefficients of the final model are shown in Table 7. From Table 7, it can be seen that the two paths from EV and GS to B are significant at the level of 0.001.

From Table 8, all the goodness-of-fit indices are within the acceptable levels, indicating the final model fits the data very well. Thus, it can be concluded that economic viability (EV) and governmental supervision (GS) are the two determinants of influencing construction professionals’ BIM adoption behavior.

Figure 4 illustrates the relationships between EV, GS, and B. It can be seen that the path weight from latent variable EV to B is 0.37, which means that when EV goes up by 1 standard deviation, the construction professionals’ BIM adoption behavior goes up by 0.37 standard deviations. Similarly, the path weight from latent variable GS to B is 0.34, meaning when GS goes up by 1 standard deviation, BIM adoption behavior goes up by 0.40 standard deviations.

## 5. Discussion

In the existing literature, studies on the influencing factors of BIM adoption have been conducted. Sun et al. [107] identified 22 influencing factors of BIM adoption from literature and classified them into five categories: technology, cost, management, personnel, and legal. However, the influencing factors of BIM adoption may vary in different countries or regions. For example, Alreshidi et al. [108] revealed that social–organizational theme, financial theme, technical theme, contractual theme, and legal theme are the five main themes of BIM adoption barriers in the UK. Ngowtanasawan [109] focused on the architectural and engineering design industry in Thailand and divided the BIM adoption factors into technology and people aspects. Hatem et al. [110] identified the motivation factors of BIM implementation in Iraq, such as contracting with international experts. Furthermore, Hatem et al. [111] revealed the barriers of BIM adoption, such as weakness of the government’s efforts, inadequate knowledge about the benefits of BIM, and resistance to change. In Hong Kong, the resistance to change by construction stakeholders was also regarded as the main barrier of BIM implementation [112]. In addition, inadequate organizational support and structure to execute BIM and lack of BIM industry standards were considered another two main barriers [112]. Ahuja et al. [113] investigated the construction market in India and categorized the BIM adoption factors into three groups, such as technological factors, organizational factors, and environmental factors.

China has a large construction market, the construction industry value in China amounted to 893.58 billion U.S. dollars in 2018 and has an increasing trend [114]. In recent years, BIM has got extensive attentions from the AEC industry and the investment in promoting BIM is expected to be increased [115]. From this survey, it can be seen that the research institutions in China have paid concentration on BIM adoption; however, the BIM adoption in practice is still insufficient, most of the respondents participated in less than five BIM projects. According to the results derived in this study, it can be concluded that, in the context of China, economic viability and governmental supervision are the two determinants for construction professionals’ BIM adoption. These two factors are also recognized in the UK investigation [108]. Nevertheless, behavioral intentions, technical feasibility, and industrial environment, which are often regarded as significant affecting factors in other countries, are insignificant in the Chinese context.

In order to verify the modeling results, three construction professionals who have rich experiences in practical BIM implementation were interviewed to collect their comments. All of the three interviewees were not surprised that behavioral intention is not a significant affecting factor for BIM adoption. They argued that making BIM adoption decision is usually organizational behavior rather than individual behavior. The individuals normally have no authority to decide using BIM or not. In terms of the technical feasibility, the three interviewees also agreed with the empirical result. They acknowledged that many technical issues were encountered during their practical implementation of BIM. For example, one interviewee claimed that though a main advantage of BIM is data interaction between different stakeholders, the contractor must rebuild the model as the architectural designer used different software in many of his previous projects. However, at present, there have been several famous localized BIM related software producers that can provide timely and effective technical support when encountering technical problems. In addition, the three interviewees believed that the current technical issues will diminished as BIM technologies develop. In terms of the industrial environment, different from the analysis results, one of the three interviewees regarded the industrial environment is a significant affecting factor for BIM adoption. He argued that BIM is a technology that involves all participants throughout the whole lifecycle of a project, and it is a must that general agreements to be achieved for BIM promotion. However, the other two interviewees claimed that the industrial environment is mainly affected by outside forces (e.g., government policies) rather than the industry itself. If there are strong outside forces that promote BIM adoption, the industrial environment will be improved automatically as the development of BIM technology. For example, suppose a project is designed aiming to obtain industrial prizes. In that case, it is usually a prior option to adopt BIM because it can help earn more scores during the assessment.

All of the three interviewees agreed that economic viability and governmental supervision are the two determinants of BIM adoption. This is echoed with the findings from Ding, Zuo, Wu and Wang [89] which investigated the architects’ perspectives. It is not difficult to understand that economic viability is a determinant because the nature of a construction company is to gain profits. The profits of adopting BIM can be obtained from cost management, schedule optimization, reduction of design change, etc. At the beginning of BIM technology development, it is unavoidable that there are many technical issues and the industrial environment is not mature; governmental supervision is important in this circumstance. Xu, Feng and Li [86] also claimed that the promotion from the government is essential to companies to adopt the BIM technology. Actually, in some cities of China (e.g., Shenzhen and Guangzhou), the government requires contractors to adopt BIM in government investment projects in order to promote BIM development.

## 6. Conclusions

Building information modeling (BIM) has received growing interest from the construction industry in recent years. Based on the theory of planned behavior, this study established an initial hypothesis model containing nine latent variables and designed a questionnaire to measure the established model. After the analysis of reliability and validity, the goodness-of-fit of the proposed model was tested by SEM. The empirical results showed that the construction professionals are willing to adopt BIM in their projects; although, the advantages of BIM cannot be realized sufficiently at this stage. The SEM analysis revealed that economic viability and governmental supervision are the critical factors of construction professionals’ BIM adoption behavior, sharing weights of 0.37 and 0.34 respectively, whereas behavioral intentions, technical feasibility, and industrial environment are not influential in this regard.

The research findings of this study have practical contributions for promoting BIM development in China. They may also be applicable in other developing countries; however, future research is suggested to be conducted in the local context. Based on the derived results, it is suggested that focus should be paid mainly on the economic viability and governmental supervision aspects to promote BIM adoption at this stage. In this regard, the government can play a significant role. Policies (e.g., awarding credits to BIM adoption during project assessment) can be issued to guide stakeholders to adopt BIM technology.

This study has some limitations. For example, the sample size collected in this study is 206, which is not so sufficient for employing SEM. Future research is suggested to collect a more significant number of responses.

## Figures and Tables

**Figure 1 ijerph-18-03022-f001:**
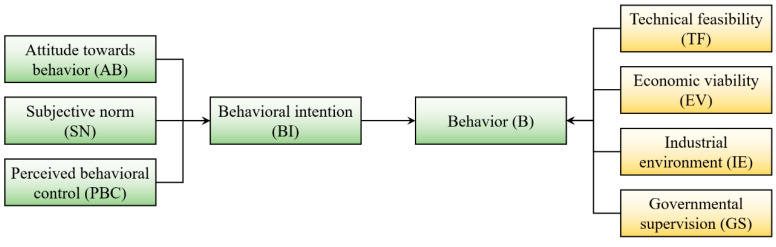
The preliminary theoretical model.

**Figure 2 ijerph-18-03022-f002:**
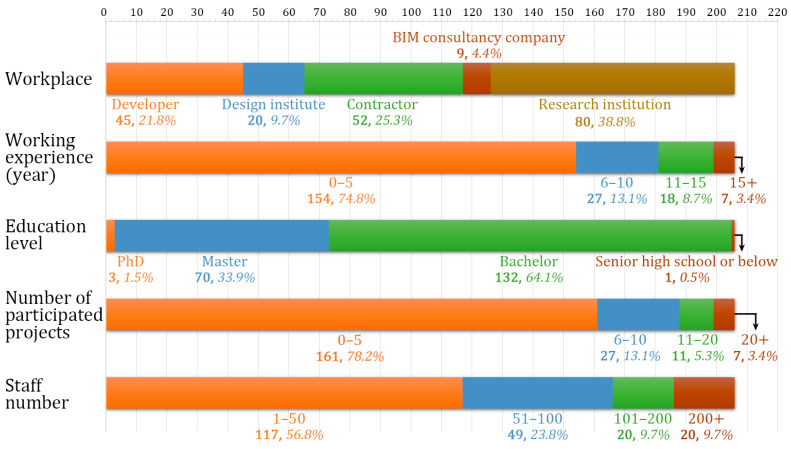
Stacked bar charts of respondents’ personal information.

**Figure 3 ijerph-18-03022-f003:**
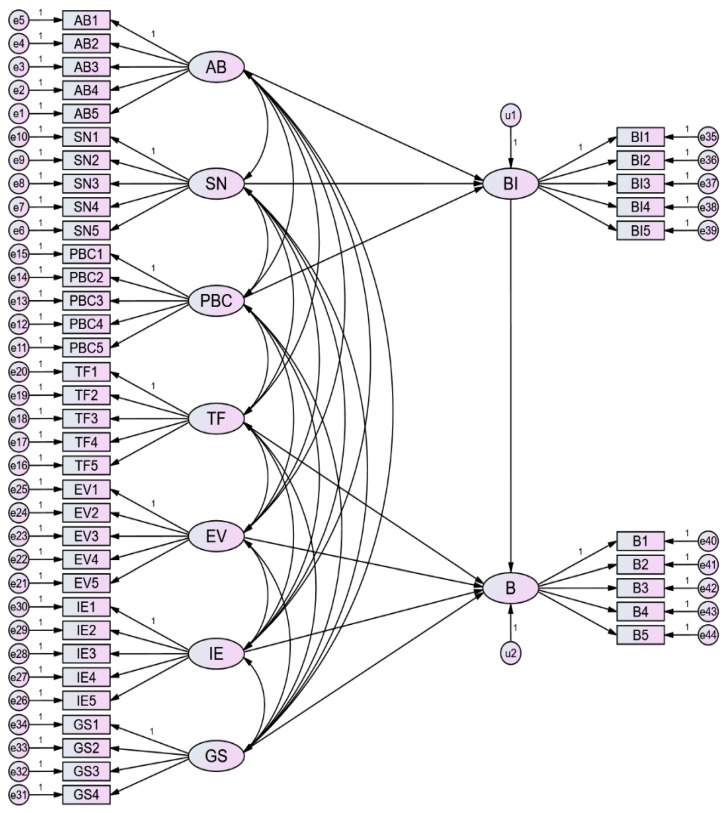
The initial structural equation model.

**Figure 4 ijerph-18-03022-f004:**
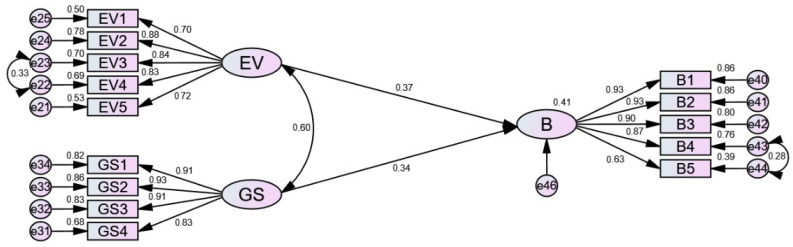
Standardized estimation of the final model.

**Table 1 ijerph-18-03022-t001:** Potential implementations and benefits of building information modeling (BIM).

Category	Item	Reference
Potential implementation	Cost management	[21,22]
	Facility management	[23,24,25,26]
	Safety management	[27]
	Green building development	[28,29,30,31,32]
	Carbon emissions calculation	[33,34,35]
	Life cycle energy efficiency	[36]
	Prefabrication	[37,38,39,40]
	Lean construction	[41,42]
	Risk management	[43,44,45]
	Energy retrofitting	[46]
	Noise mitigation	[47]
Benefit	Optimizing design solutions	[48,49,50]
	Enhancing visualization	[51,52,53]
	Improving teamwork	[54,55]
	Increasing productivity	[24,56,57]
	Saving time and expense	[58,59,60]
	Reducing waste	[61,62,63]
	Lifecycle management	[54,64,65]

**Table 2 ijerph-18-03022-t002:** Measurement scales in the formal questionnaire.

Latent Variable	Measurement Item	Measurement Scale
Attitude towards behavior (AB)	AB1	I think work efficiency can be improved by using BIM.
	AB2	I think construction period can be shorten by using BIM.
	AB3	I think project life cycle cost can be reduced by using BIM.
	AB4	I think the quality of the project can improved by using BIM.
	AB5	I think the image of the project can enhanced by using BIM.
Subjective norm (SN)	SN1	My superior thinks that mastering BIM is helpful to my potential career development.
	SN2	My colleagues could approve me better if I am skilled at using BIM.
	SN3	My family supports me to use BIM in my project.
	SN4	The developer expects me use BIM in my project.
	SN5	The local government encourages me to use BIM in my project.
Perceived behavioral control (PBC)	PBC1	I have enough opportunities to use BIM in my project.
	PBC2	I can get enough support to use BIM in my project.
	PBC3	I have enough time to use BIM in my project.
	PBC4	I have enough experience to use BIM in my project.
	PBC5	I have adequate equipment and software to use BIM in my project.
Behavioral intention (BI)	BI1	I am willing to use BIM technology to demonstrate the project.
	BI2	I am willing to use BIM technology for design optimization during the project design phase.
	BI3	I am willing to use BIM technology for project management in the construction process.
	BI4	I am willing to learn and to use new applications of BIM technology.
	BI5	I am willing to participate in BIM training.
Technical feasibility (TF)	TF1	Data are compatible amongst different existing BIM software.
	TF2	The localized BIM software has been developed in China.
	TF3	There have been reliable platforms for BIM data exchange.
	TF4	The existing BIM software has good potential for function extension.
	TF5	Technical support can be received in the process of using BIM.
Economic viability (EV)	EV1	Enterprise can receive satisfactory returns from using BIM.
	EV2	Enterprise has sufficient funding for purchasing BIM related equipment and software.
	EV3	Enterprise has sufficient funding for BIM related consultancy.
	EV4	Enterprise has sufficient funding for training BIM employees.
	EV5	Government has attractive incentives for promoting BIM adoption.
Industrial environment (IE)	IE1	There is a generic BIM standard in the industry.
	IE2	There is a generic BIM contract template in the industry.
	IE3	There are good communications between different project stakeholders.
	IE4	The construction professionals are willing to learn and to use BIM.
	IE5	There are sufficient successful BIM practices in the industry.
Governmental supervision (GS)	GS1	There have been regulations for protecting BIM related intellectual property rights.
	GS2	There have been regulations for protecting the benefits of different stakeholders in a BIM project.
	GS3	There have been disputation resolution mechanisms for BIM projects.
	GS4	There has been a specific government department to supervise BIM implementation in projects.
Behavior (B)	B1	I use BIM to improve work efficiency in the project.
	B2	I use BIM to optimize design in the project.
	B3	I use BIM for cross-disciplinary work coordination in the project.
	B4	I use BIM to demonstrate the project.
	B5	I have participated in BIM related workshop or training.

**Table 3 ijerph-18-03022-t003:** Reliability and validity analysis.

Latent Variables	Items	Factor Loadings	Cronbach’s Alpha	KMO Measure	Bartlett’s Test of Sphericity		
					Chi-Square	df	Sig.
AB	AB1	0.853	0.901	0.875	606.650	10	0.000
	AB2	0.864					
	AB3	0.861					
	AB4	0.861					
	AB5	0.794					
SN	SN1	0.886	0.896	0.818	665.012	10	0.000
	SN2	0.897					
	SN3	0.840					
	SN4	0.790					
	SN5	0.787					
PBC	PBC1	0.886	0.927	0.878	806.143	10	0.000
	PBC2	0.906					
	PBC3	0.901					
	PBC4	0.843					
	PBC5	0.864					
BI	BI1	0.865	0.925	0.893	762.501	10	0.000
	BI2	0.900					
	BI3	0.857					
	BI4	0.902					
	BI5	0.865					
TF	TF1	0.807	0.896	0.870	600.738	10	0.000
	TF2	0.830					
	TF3	0.914					
	TF4	0.834					
	TF5	0.819					
EV	EV1	0.764	0.899	0.853	648.717	10	0.000
	EV2	0.894					
	EV3	0.878					
	EV4	0.889					
	EV5	0.793					
IE	IE1	0.876	0.894	0.797	665.143	10	0.000
	IE2	0.874					
	IE3	0.858					
	IE4	0.762					
	IE5	0.817					
GS	GS1	0.924	0.940	0.854	758.238	6	0.000
	GS2	0.943					
	GS3	0.934					
	GS4	0.881					
B	B1	0.922	0.932	0.883	923.902	10	0.000
	B2	0.923					
	B3	0.916					
	B4	0.919					
	B5	0.743					

**Table 4 ijerph-18-03022-t004:** Path Coefficient of the initial model.

Model Path	Estimate	S.E.	C.R.	*p*
BI ← AB	0.340	0.091	3.733	***
BI ← SN	0.299	0.087	3.449	***
BI ← PBC	0.061	0.055	1.116	0.264
B ← TF	0.037	0.195	0.189	0.850
B ← EV	0.765	0.232	3.301	***
B ← IE	0.086	0.214	0.403	0.687
B ← GS	0.336	0.160	2.099	0.036
B ← BI	0.200	0.126	1.580	0.114
AB5 ← AB	1.064	0.090	11.867	***
AB4 ← AB	1.238	0.094	13.203	***
AB3 ← AB	1.119	0.084	13.240	***
AB2 ← AB	1.118	0.083	13.547	***
AB1 ← AB	1.000			
SN5 ← SN	0.703	0.061	11.511	***
SN4 ← SN	0.729	0.061	11.938	***
SN3 ← SN	0.918	0.062	14.696	***
SN2 ← SN	0.954	0.053	18.018	***
SN1 ← SN	1.000			
PBC5 ← PBC	0.954	0.065	14.578	***
PBC4 ← PBC	0.942	0.068	13.935	***
PBC3 ← PBC	1.011	0.057	17.723	***
PBC2 ← PBC	0.980	0.054	18.290	***
PBC1 ← PBC	1.000			
TF5 ← TF	0.898	0.079	11.370	***
TF4 ← TF	0.906	0.080	11.331	***
TF3 ← TF	1.189	0.088	13.514	***
TF2 ← TF	1.027	0.087	11.735	***
TF1 ← TF	1.000			
EV5 ← EV	1.085	0.108	10.044	***
EV4 ← EV	1.376	0.120	11.464	***
EV3 ← EV	1.309	0.113	11.571	***
EV2 ← EV	1.297	0.108	11.977	***
EV1 ← EV	1.000			
IE5 ← IE	0.758	0.063	12.059	***
IE4 ← IE	0.642	0.062	10.290	***
IE3 ← IE	0.861	0.059	14.505	***
IE2 ← IE	0.966	0.053	18.265	***
IE1 ← IE	1.000			
GS4 ← GS	0.831	0.050	16.546	***
GS3 ← GS	1.006	0.047	21.298	***
GS2 ← GS	0.934	0.043	21.875	***
GS1 ← GS	1.000			
BI5 ← BI	1.029	0.067	14.170	***
BI4 ← BI	0.996	0.068	15.367	***
BI3 ← BI	0.946	0.065	13.914	***
BI2 ← BI	1.047	0.073	15.626	***
BI1 ← BI	1.000			
B5 ← B	0.646	0.058	11.056	***
B4 ← B	0.994	0.050	19.744	***
B3 ← B	1.054	0.050	21.150	***
B2 ← B	1.057	0.045	23.354	***
B1 ← B	1.000			

Note: *** Statistically significant at the 0.001 level of confidence.

**Table 5 ijerph-18-03022-t005:** Construct reliability (CR) values and the average of variance extracted (AVE) values of the initial model.

Latent Variables	Items	Standardized Factor Load Estimation	CR Values	AVE Values	Judgment
AB	AB1	0.818	0.9015	0.647	√
	AB2	0.818			
	AB3	0.807			
	AB4	0.822			
	AB5	0.755			
SN	SN1	0.888	0.8962	0.6369	√
	SN2	0.902			
	SN3	0.814			
	SN4	0.689			
	SN5	0.667			
PBC	PBC1	0.882	0.9245	0.711	√
	PBC2	0.897			
	PBC3	0.885			
	PBC4	0.760			
	PBC5	0.782			
BI	BI1	0.828	0.9262	0.7154	√
	BI2	0.879			
	BI3	0.812			
	BI4	0.876			
	BI5	0.832			
TF	TF1	0.755	0.8992	0.6419	√
	TF2	0.795			
	TF3	0.903			
	TF4	0.770			
	TF5	0.774			
EV	EV1	0.716	0.9031	0.6525	√
	EV2	0.872			
	EV3	0.858			
	EV4	0.853			
	EV5	0.725			
IE	IE1	0.808	0.8869	0.6117	√
	IE2	0.818			
	IE3	0.828			
	IE4	0.679			
	IE5	0.768			
GS	GS1	0.905	0.9413	0.8008	√
	GS2	0.925			
	GS3	0.919			
	GS4	0.827			
B	B1	0.922	0.9335	0.7402	√
	B2	0.923			
	B3	0.900			
	B4	0.877			
	B5	0.648			

Note: √ represent the acceptable threshold.

**Table 6 ijerph-18-03022-t006:** Goodness-of-fit of the initial model.

Goodness-of-Fit Measure		Level of Acceptance Fit	Fit Statistics	Judgment
Absolute fit	χ^2^/df	<5 acceptable; <3 good	1.805	√
	GFI	>0.8 acceptable; >0.9 good	0.817	√
	AGFI	>0.8 acceptable; >0.9 good	0.777	×
	RMSEA	<0.1 acceptable; <0.08 good	0.062	√
Incremental fit	NFI	>0.9	0.84	×
	RFI	>0.9	0.816	×
	IFI	>0.9	0.925	√
	TLI	>0.9	0.91	√
	CFI	>0.9	0.932	√

Note: √ represent the acceptable threshold; × represent the unacceptable threshold.

**Table 7 ijerph-18-03022-t007:** Path Coefficient of the final model.

Model Path	Estimate	S.E.	C.R.	*p*
B ← EV	0.696	0.155	4.502	***
B ← GS	0.421	0.096	4.401	***
EV5 ← EV	1.100	0.113	9.732	***
EV4 ← EV	1.363	0.127	10.758	***
EV3 ← EV	1.294	0.119	10.843	***
EV2 ← EV	1.331	0.114	11.640	***
EV1 ← EV	1.000			
GS4 ← GS	0.825	0.050	16.514	***
GS3 ← GS	0.996	0.047	21.202	***
GS2 ← GS	0.931	0.042	22.110	***
GS1 ← GS	1.000			
B5 ← B	0.622	0.059	10.588	***
B4 ← B	0.980	0.050	19.496	***
B3 ← B	1.046	0.049	21.175	***
B2 ← B	1.056	0.044	23.864	***
B1 ← B	1.000			

Note: *** Statistically significant at the 0.001 level of confidence.

**Table 8 ijerph-18-03022-t008:** Goodness-of-fit of the final model.

Goodness-of-Fit Measure		Level of Acceptance Fit	Fit Statistics	Judgment
Absolute fit	χ^2^/df	<5 acceptable; <3 good	2.328	√
	GFI	>0.8 acceptable; >0.9 good	0.921	√
	AGFI	>0.8 acceptable; >0.9 good	0.893	√
	RMSEA	<0.1 acceptable; <0.08 good	0.069	√
Incremental fit	NFI	>0.9	0.932	√
	RFI	>0.9	0.914	√
	IFI	>0.9	0.958	√
	TLI	>0.9	0.946	√
	CFI	>0.9	0.957	√

Note: √ represent the acceptable threshold.

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
