# Peer review of "Investigating the Critical Factors of Professionals’ BIM Adoption Behavior Based on the Theory of Planned Behavior"

_ijerph, 2021, doi:10.3390/ijerph18063022_

Round 1

Reviewer 1 Report

Presenting the most critical factors influencing the BIM adoption behavior of construction professionals in China is important for expanding the use of BIM. However, in order to emphasize the contribution of the article, and for the purpose of understanding its usefulness, authors are advised to refer to the following comments

  1. General comments

Corrections are required regarding grammar, and the editing of sentences. It is important to use the right words for accurate presentation. If the text is improved, the message you intend to convey will be much clearer and more understandable. Some examples that require appropriate consideration are presented below.

  1. Lines 64-65. Please correct the wording of the sentence to show a logical connection between the words.
  2. Lines 105, 122. Please check if the acronyms need to be repeated again.
  3. Lines 321-324. The sentence needs to be reworded through its division, for more understanding.
  4. Line 346. Please use a more appropriate term than "company behavior" or, alternatively, explain this concept. Is not the decision to adopt a technology by a company the behavior of the individuals who run it?
  5. Lines 375-377. The sentence needs to be reworded through its division, for more understanding.

2. Specific comments

2.1. Introduction

  1. The authors note that BIM has shown great potential for problem solving in the traditional architecture, engineering and construction industry. However, despite the many benefits of BIM, the benefits have not been sufficiently achieved in practice. Under these circumstances, the authors argue, there is value in systematically exploring the driving factors of BIM adoption in the Chinese context. It is not clear how examining the lack of implementation of BIM benefits relates to BIM adoption in the industry. Does BIM adoption guarantee full application of its benefits?
  2. The authors note that it is appropriate for their study to implement the TPB as the basic theoretical framework. The justification for choosing TPB as a theoretical framework should be noted. Why is it appropriate? What can it add to other studies in relation to BIM application in industry?

2.2. Research methodology

  1. The authors note that three sections were involved in the questionnaire. It is not clear where the questions in the measurement scales section and in the open questions section were taken from.
  2. It is necessary to explain how the sections are combined to reach the conclusions.

2.3. Discussions

  1. The authors note that the factors influencing BIM adoption may vary between different countries or regions. It is important to explain the characteristics of the Chinese industry, in order to distinguish or apply the findings in other industries.
  2. It is important to address the characteristics of the respondents, thus connecting all the methodological sections. It is advisable to detail how the characteristics of the respondents were reflected in the answers. Did the large number of respondents from among a research institution influence the results? Is it important that most of the respondents participated in a small amount of projects in the industry?

2.4. Conclusions

  1. It is important to address the practical significance of the conclusions in China and in relation to the application of research insights to other industries as well.
  2. It should be detailed how the results may help to realize the potential of BIM and what are the tools offered by the authors to realize the conclusions.

Author Response

We appreciate the constructive comments and suggestions from the Reviewer #1. We believe that the quality of this manuscript has been significantly improved after the revisions according to the reviewer’s comments. Please kindly find the attached file for our detailed point-to-point responses.

Reviewer 2 Report

The paper discusses the factors impacting the adoption of BIM in China. 

The methodology used is relevant but the number of the valid responses from the survey (on which the whole statistical analysis is based) is barely sufficient. 

Some other remarks:

  • Some references are quite old (given the recent BIM timeline): 
    "For example, Cao, et al. [72] investigated 106 real-life projects in 80 China, revealing that BIM was principally employed as a visualization tool; the other main 81 advantages of BIM were rarely achieved in the current AEC industry"
  • line 89: driving factors (instead of 'driven')
  • line 110: review the phrase (the English)
  • in the table - p.6, TF2 : reformulate the phrase : There has been localized BIM software in China
  • in the same table: B1-to B5 - all the corresponding phrases should be checked (the tense of the verb seems inappropriate)

What % of all targeted companies represent the 206 valid responses? Is this a representative sample for the Chinese construction industry?

Author Response

We appreciate the constructive comments and suggestions from the Reviewer #2. We believe that the quality of this manuscript has been significantly improved after the revisions according to the reviewer’s comments. Please kindly find the attached file for our detailed point-to-point responses.

Round 2

Reviewer 1 Report

The authors addressed the comments

Reviewer 2 Report

The article is substantially ameliorated.